# Immunogenicity and Efficacy of a Recombinant Human Adenovirus Type 5 Vaccine against Zika Virus

**DOI:** 10.3390/vaccines8020170

**Published:** 2020-04-07

**Authors:** Tara Steffen, Mariah Hassert, Stella G. Hoft, E. Taylor Stone, Jianfeng Zhang, Elizabeth Geerling, Brian T. Grimberg, M. Scot Roberts, Amelia K. Pinto, James D. Brien

**Affiliations:** 1Department of Molecular Microbiology and Immunology, Saint Louis University, St Louis, MO 63103, USA; tara.steffen@slu.edu (T.S.); mariah.hassert@slu.edu (M.H.); stella.hoft@health.slu.edu (S.G.H.); taylor.t.stone@slu.edu (E.T.S.); lizzie.geerling@slu.edu (E.G.); 2Altimmune, Inc., Gaithersburg, MD 20878, USA; jzhang@altimmune.com (J.Z.); sroberts@altimmune.com (M.S.R.); 3Center for Global Health and Diseases, Department of Pathology, Case Western Reserve University School of Medicine, Cleveland, OH 44106, USA; brian.grimberg@case.edu

**Keywords:** Zika virus vaccine, pre-clinical development, immune response, animal model

## Abstract

Zika virus (ZIKV) is a significant public health concern due to the pathogen’s ability to be transmitted by either mosquito bite or sexual transmission, allowing spread to occur throughout the world. The potential consequences of ZIKV infection to human health, specifically neonates, necessitates the development of a safe and effective Zika virus vaccine. Here, we developed an intranasal Zika vaccine based upon the replication-deficient human adenovirus serotype 5 (hAd5) expressing ZIKV pre-membrane and envelope protein (hAd5-ZKV). The hAd5-ZKV vaccine is able to induce both cell-mediated and humoral immune responses to ZIKV epitopes. Importantly, this vaccine generated CD8^+^ T cells specific for a dominant ZIKV T cell epitope and is shown to be protective against a ZIKV challenge by using a pre-clinical model of ZIKV disease. We also demonstrate that the vaccine expresses pre-membrane and envelope protein in a confirmation recognized by ZIKV experienced individuals. Our studies demonstrate that this adenovirus-based vaccine expressing ZIKV proteins is immunogenic and protective in mice, and it encodes ZIKV proteins in a conformation recognized by the human antibody repertoire.

## 1. Introduction

Zika virus (ZIKV) is a re-emerging arbovirus of the *Flavivirus* genus that includes multiple important human pathogens including dengue viruses 1–4 (DENV), Japanese encephalitis virus (JEV), West Nile virus (WNV), and yellow fever virus (YFV) [1]. Similar to other flaviviruses in this family, ZIKV transmission occurs from the bite of an infected *Aedes aegypti* mosquito [2]. One distinctive feature of ZIKV is its ability to be transmitted horizontally through sexual contact [3] and vertically from mother to developing fetus [4]. 

The development of a ZIKV vaccine is urgent due to the speed of ZIKV spread in susceptible populations, as well as the range of diseases caused by infection [5]. According to the World Health Organization (WHO), a majority of individuals who are infected with ZIKV are asymptomatic, but those with a mild clinical form of the disease report symptoms including fever, rash, conjunctivitis, and muscle and joint pain, all of which typically last less than a week. While these symptoms are rarely debilitating, complications of Zika virus disease can include Guillain–Barré syndrome. Zika-induced microcephaly and congenital abnormalities are generally classified as congenital Zika syndrome (CZS), which is also a potential complication in the developing fetuses of pregnant women [4,6,7,8,9,10]. These risks, combined with the explosive outbreaks that occurred at Yap Island in 2007 [11], French Polynesia in 2013 [12], and Brazil in 2015–16 [13], underscore the need to control the spread of ZIKV. However, there are no current FDA-approved vaccines available to control infection. 

Harnessing the power of both arms of the adaptive immune response improves vaccine potency and efficacy while reducing opportunities of viral escape. Current data from both naturally acquired human infection and mouse models of ZIKV disease indicate that protection from disease is multi-faceted (reviewed in [14,15,16,17,18]). Both small and large animal models of ZIKV infection have demonstrated that the Zika-specific CD8^+^ and CD4^+^ T cell response is necessary and sufficient to protect against a lethal ZIKV challenge [19,20,21,22]. As has been demonstrated previously for multiple flavivirus infections [23,24,25], ZIKV-specific polyclonal and monoclonal neutralizing antibody responses can protect mice and nonhuman primates from mortality and severe disease. There is also a role for non-neutralizing antibodies to protect against ZIKV disease [22,26,27,28]. These data highlight the necessity for a ZIKV vaccine that induces robust T cell and B cell responses for protection against disease.

Successful flavivirus vaccines, including those developed against YFV, JEV, and tick-borne encephalitis virus (TBEV), all express structural flavivirus antigens (reviewed in [29,30,31]). As neutralizing antibody responses are predominately directed toward structural proteins, structural antigens are desired candidates for incorporation into vaccines because of the potential for both T and B cell epitopes to be expressed (reviewed in [32]). Therefore, similar to other flavivirus vaccines, it is assumed that an effective ZIKV vaccine should express some, if not all, of the viral structural proteins. The 10.7 Kb ZIKV RNA genome encodes three structural proteins: capsid (C), pre-membrane (prM), and envelope (E), as well as seven non-structural proteins that include NS1, NS2A, NS2B, NS3, NS4a, NS4b, and NS5 [33]. For the structural proteins, cryogenic-electron microscopy (cryo-EM), and other structural studies have shown that the C protein is found complexed with the RNA genome, which is encapsulated by 180 copies of both the prM and E glycoproteins with quasi-icosahedral symmetry [34,35,36]. The flavivirus E protein is comprised of three domains (DI, DII, and DIII) and is essential for viral receptor recognition and viral fusion [34,37,38]. The presence of these three structural proteins on the virus surface make them all possible antibody epitopes. Studies done to map the antibody response to ZIKV have shown that, similar to other flaviviruses, the bulk of the antibody response is directed toward the E protein [39,40,41]. Additionally, evidence from natural history studies indicate that ZIKV E stimulates both a T cell and antibody response in humans [42,43,44,45,46]. Therefore, a ZIKV vaccine that expresses the ZIKV E protein would be anticipated to induce antibodies and antigen-specific anti-viral T cells.

The hallmarks of an effective vaccine are the ability to express viral antigens at a significant level, to induce immunogenicity, and to protect the host from severe disease associated with infection. Here, we developed an E1A/E3-deleted, adenovirus-vectored vaccine against ZIKV. Adenovirus-based vaccines have been shown to safely elicit protective responses to other pathogens [47]. In this study, we investigated the potential for a protective and immunogenic human adenovirus serotype 5 (hAd5)-based ZIKV vaccine based on the structural proteins prM and E with an E209A amino acid mutation that has been reported to improve the processing of the prM-E polypeptide [48]. Here, we show that our vaccine construct, hAd5-ZKV, induces a CD8^+^ T cell response and humoral response against known ZIKV antigens. We then demonstrate that a single intranasal vaccine administration is protective from ZIKV-induced weight loss and mortality with a pre-clinical model of ZIKV disease. Finally, we demonstrate that the hAd5-ZKV vaccine expresses E antigens that are recognized by the human antibody response generated during natural ZIKV infection. These results provide strong evidence that an adenovirus-vectored vaccine against ZIKV can induce a potent protective immune response against this emerging pathogen threat.

## 2. Materials and Methods

### 2.1. Ethics Statement

This study was carried out in accordance with the recommendations in the Guide for the Care and Use of Laboratory Animals of the National Institutes of Health. The protocols were approved by the Institutional Animal Care and Use Committee at the Saint Louis University (Assurance Number: D16–00141).

### 2.2. Viruses, Human Serum and Cells

The ZIKV strain PRVABC59 used was a kind gift of Robert Lanciotti (CDC). The ZIKV used was grown in Vero cells from the American Type Culture Collection (ATCC CCL-81) and stored at −80 °C, as described previously [19,22,49]. The human serum samples were obtained from the University of North Carolina/ National Institute of Health (UNC/NIH) Traveler Study through BEI Resources, NIAID, NIH: human convalescent plasma.

### 2.3. Zika Virus Sequence Analysis

Full length prM and E sequences derived from human isolates were identified and downloaded from Virus Variation Resource [50]. prM and E sequences were trimmed and aligned with Geneious 7. Single amino acid variants were identified and quantified using the analyze sequence variation tool of the NIAID Virus Pathogen Database and Analysis Resource (VIPR) [51]. The resulting amino acid changes were mapped onto the structural genes using a plot protein [52].

### 2.4. Recombinant Replication Deficient Adenovirus Type 5

Replication-deficient human adenovirus 5 (hAd5) were generated using previously described methods [53]. In brief, the hAd5 vector lacks the essential E1 gene, rendering it replication-deficient. The deletion of the E3 genes allows for additional genomic space for the transgene cassette. The expression cassette is a cytomegalovirus (CMV)-immediate, early-driven transgene encoding a tissue plasminogen activator signal sequence (tPA) followed by a human-codon-optimized ZIKV prM-E cassette inserted into the E1 region of the adenovirus vector. The ZIKV prM-E insert encodes amino acid 126–794, which includes the full length pre-membrane; the full length envelope protein including domains I, II, and III; and the stem anchor domains, in addition to harboring an E209A mutation that has been reported to improve the processing of the prM-E polypeptide [48]. Vectors and virus were grown on Human Embryonic Kidney 293 cells (HEK293) and purified by two successive banding of the vector in a CsCl step gradient followed by desalting and storage at −80 °C.

### 2.5. Focus Forming Assay

ZIKV and hAd5-ZKV titration was performed with a focus forming assay (FFA), as described previously [49,54]. Briefly, Vero cells were plated the day before the assay in a 96-well plate. Virus stocks were then serially diluted, allowed to infect cells, and overlaid with methylcellulose. The cells were incubated at 37 °C for 48 h, followed by fixation with paraformaldehyde. Immunostaining with the pan-flavivirus monoclonal antibody 4G2 (ATCC) and goat anti-mouse secondary (Molecular Probes) was used to visualize the formation of the foci resulting from either the ZIKV or hAd5-ZKV stock.

### 2.6. Western Blot

To confirm protein expression from the recombinant hAd5 virus, 293 cells were infected with hAd5-ZKV at a multiplicity of infection (MOI) of 5. After 48 h of incubation at 37 °C, 5% CO2, the cells were rinsed once with PBS and harvested with a 2× NuPage buffer. Samples were heated at 95 °C for 10 min, cooled, and loaded onto NuPage 4%–12% gels. Protein was transferred to nitrocellulose membrane, blocked with non-fat milk, and probed with primary antibodies (4G2 and GeneTex anti-prM (GTX-133305) Irvine, USA) in TBST. After overnight incubation, membranes were washed 3 times with TBST and blotted with fluorescently-conjugated secondary antibodies (molecular probes). After one hour in secondary, membranes were washed three times in TBST and one time in PBS, and then the fluorescent signal was captured with an Amersham Typhoon imager.

### 2.7. Antigen-Specific T Cell Staining

Spleens were harvested from vaccinated WT C57BL/6 mice eight days post vaccination. Spleens were ground over a 100 μm cell strainer and brought up in Roswell Park Memorial Institute (RPMI) 1640 Medium with 10% FBS and HEPES ((4-(2-hydroxyethyl)-1-piperazineethanesulfonic acid)). Then, ~10^6^ cells were plated per well in a round bottom 96-well plate and stimulated for 6 h at 37 °C, 5% CO_2_ in the presence of 10 μg/mL brefeldin A and either α-CD3 (2C11 clone) or 10 μg of peptide in 90% DMSO. Following peptide stimulation, cells were washed once with PBS and stained for the following surface markers: α-CD8-PerCP-Cy 5.5 (clone 53–6.7), α-CD3-AF700 (clone 500A2), and α-CD19-BV605 (clone 1D3). Cells were then fixed and permeabilized and stained for the following intracellular marker: α-IFN-γ-APC (clone B27). The cells were analyzed by flow cytometry using an Attune-NXT.

### 2.8. ELISA

Polystyrene 96-well plates were coated overnight at 4 °C with 1 µg/mL of ZIKV E protein (Sinobiological) in a sodium carbonate buffer (pH 9.3). Plates were washed three times in PBS with 0.02% Tween 20 and blocked with non-fat dried milk for one hour at 37 °C with PBS, 2% BSA, and 0.02% Tween 20. Serum from hAd5-ZKV-vaccinated mice was serially diluted in PBS and then incubated at 37 °C. Plates were washed four times with PBS containing 0.02% Tween 20 and incubated with an biotin-labeled goat anti-mouse secondary antibody (Jackson Labs) for one hour. After washing and incubation with horseradish peroxidase (HRP)-conjugated streptavidin (Vector Laboratories, Burlingame, USA), plates were developed with tetramethylbenzidine substrate (Dako). The reaction was stopped with the addition of 2 N H_2_SO_4_, and emission (450 nm) was read with a microplate reader (Molecular Devices, San Jose, USA).

### 2.9. Vaccination with Recombinant Replication Deficient Adenovirus Type 5

Wild type C57BL/6 and *Ifnar1*^-/-^ mice homozygous for interferon alpha and beta receptor subunit knockout mutation on the C57BL/6J background were purchased from Jackson Laboratory and bred at Saint Louis University School of Medicine. Mice were anesthetized with ketamine/xylazine (90 mg/kg: 10 mg/kg), then intranasally (i.n.) vaccinated with 20 ul of hAd5-ZKV diluted in PBS at a concentration of 1.5 × 10^9^ particles per ml.

### 2.10. Animal Challenge

*Ifnar1*^-/-^ mice homozygous for interferon alpha and beta receptor subunit knockout mutation on the C57BL/6J background were ordered from the Jackson Laboratory and bred at the Saint Louis University School of Medicine. ZIKV was diluted in sterile PBS at pH 7.4 to obtain a final concentration of 1 × 10^4^ focus forming units (FFU) of ZIKV (PRVABC59) per mouse in a final volume of 100 µL. Virus challenge was performed by i.v. injection 21 days post hAd5-ZKV vaccination. Mice were anesthetized during this procedure using ketamine/xylazine (90 mg/kg: 10 mg/kg). After challenge, each mouse was examined for visible trauma and placed back into its cage for recovery.

### 2.11. Clinical Monitoring

Animals were observed for clinical outcomes daily after ZIKV challenge. Body weight changes and mortality were recorded for each animal daily. Animal activity score assessments from 0 to 6 were recorded for each animal as follows: Paralysis Scoring: 0 = clinically normal; 1 = tail tone, flaccid tail; 2 = hind limb weakness, grasping, and balance; 3 = hind limb paraplegia, inability to move hind limbs, or urinary incontinence; 4 = weakness of fore limbs in addition to hind limb dysfunction and or distended abdomen; 5 = quadriplegia, inability to move hind limbs and forelimbs; and 6 = moribund, animal is not moving.

### 2.12. Statistical Analysis

Statistical tests for each experiment were performed using Graphpad Prism versions 6 and 8. For the evaluation of CD8^+^ responses, a Mann–Whitney test was completed to determine statistical differences. Differences in antibody binding by ELISA were determined using a Student’s two tailed t test. The protective capacity of the vaccine, as measured by animal survival, was determined using a log-rank (Mantel–Cox) test.

## 3. Results

### 3.1. Generation of Recombinant Human Adenovirus Type 5 ZIKV Vaccine

To develop a human adenovirus type 5 ZIKV vaccine (hAd5-ZKV), we first generated an expression cassette containing the CMV promoter and the tissue plasminogen activator (tPA) leader sequence (Figure 1A). The tPA sequence was followed by the human, codon-optimized, full length ZIKV prM and E proteins (amino acids 123–794) from the 2015 ZIKV isolate from Puerto Rico (PRVABC59). Within the prM protein, we utilized the E209***A*** mutation to improve prM cleavage [48] (Figure 1B). The expression cassette containing the CMV promoter and tPA-ZIKV coding sequence was inserted into the E1 region of the hAd5 vector where E1 and E3 had been deleted. The absence of the E1 gene renders the hAd5 replication-deficient, and the removal of E3 prevents the multiple immune evasion functions encoded by adenoviruses (reviewed in [55]). The completed hAd5-ZKV vaccine candidate was therefore designed to optimally express the immunogenic prM and E proteins.

To determine the potential antigenic variation encoded by the hAd5-ZKV vaccine, we identified 251 ZIKV isolated from humans with full length prM and E sequences within NCBI. Of the 251 sequences, two were removed because they had less than 50% sequence identity and appeared to contain a large number of sequencing errors. The remaining 249 sequences were translated and then aligned, which resulted in a multiple sequence alignment of 672 amino acids (aa) with 81.7% identical sites and a pairwise identity of 99.7% (Appendix A). The prevalence of aa variation was determined with an analyze sequence variation tool (viprbrc) [51], and this variation is displayed by a lollipop separately for prM and E (Figure 1C) [52]. As shown by the lollipop plots, there were four sites of aa variation—the locations are marked by blue dots—within prM with three sites within the ectodomain having a low level of aa conservation, as illustrated by the dip in the conservation plot. The E protein had 18 sites of aa variation. At 17 sites, the variation was minimal, with 1–3 of the 249 sequences differing from the vaccine aa sequence. With position 418 within the TM domain having more variation, with 8 out of the 249 sequences having aa that differed from the vaccine, accounting for a low level of conservation among the 249 sequences identified. Overall, the majority of sites of variation in E and prM that are highlighted by blue dots had such highly conserved identity across all 249 sequences that they were not fully able to be visualized by a dip in the conservation score plot. The low level of the aa sequence variation of prM and E in human ZIKV isolates supported the use of these highly sequence conserved antigens within the hAd5-ZKV vaccine.

We next set out to investigate the expression level of ZIKV proteins from the hAd5-ZKV vaccine by comparing the expression of prM and E proteins between ZIKV-infected Vero cells and hAd5-ZKV-infected 293 cells via Western blot. To determine if the hAd5-ZKV expressed prM, we probed the Western blot with an anti-ZIKV rabbit polyclonal specific to prM (Figure 1D). The detection of bands on the Western blot from the hAd5-ZKV-293-infected lysates demonstrated that the hAd5-ZKV vaccine expresses prM. However, we were surprised to note that the prM expressed from the hAd5-ZKV migrated more slowly than the prM detected from the ZIKV-infected Vero cells. This could have been due to a poor cleavage of the tPA signal sequence or the reduced cleavage of the pr peptide by furin, affecting prM folding. To determine this, further studies to analyze the exact nature of the size difference between the prM expressed from the hAd5-ZKV- and the ZIKV-infected Vero cells will need to be conducted. To evaluate the expression of E from hAd5-ZKV, 293 cells were infected with either ZIKV or hAd5-ZKV, and E protein expression was assessed by Western blot using the flavivirus cross-reactive anti E monoclonal D1–4G2–4-15 [55] (Figure 1E). As previously shown with prM, the hAd5-ZKV was able to express the ZIKV E protein, and, in the case of ZIKV E, we detected a similar migration pattern between the Vero-infected cells and the hAd5-ZKV. Our Western blots were therefore able to confirm that the hAd5-ZKV was able to express both ZIKV prM and E.

### 3.2. Immunogenicity of hAd5-ZKV Vaccine

After demonstrating that the hAd5-ZKV was able to express both prM and E, we next set out to determine if the expressed proteins could induce a potent detectible immune response. Traditionally, the immunogenicity of a vaccine is only assessed after immunization followed by one or more boosts. Given the robust expression of prM and E that we observed in our Western blot studies, we were curious to see if we could identify CD8^+^ and antibody responses in our mouse model following only a single immunization with the hAd5-ZKV vaccine candidate. The immunogenicity of the hAd5-ZKV vaccine was assessed with two independent cohorts of eight-week-old wild type C57BL/6 mice immunized i.n. with 3 × 10^7^ viral particles, as shown in Figure 2A. Mice were vaccinated intranasally, which is a potential route for clinical evaluation used to bypass pre-existing immunity to hAd5 in both animal models and in humans [56].

To assess the ability of the hAd5-ZKV vaccine to induce a CD8^+^ T cell response eight days post immunization with the hAd5-ZKV, a subset of vaccinated mice was euthanized and ZIKV antigen-specific CD8^+^ T cell responses were evaluated by intracellular cytokine staining. Splenic CD8^+^ T cells were stimulated with the immunodominant ZIKV peptide E_294_ (aa294–302) [19], and the intracellular levels of the effector cytokine interferon gamma (IFNγ were determined. A representative flow plot with our gating strategy for the detection of the of IFNγ-CD8^+^ T cells is shown in Appendix A. The total number of CD19^-^, CD3^+^, CD8^+^, and IFN-γ^+^ cells from both the vaccinated and control mice were enumerated (Figure 2C). The results from this study showed that the vaccinated animals had an average of 0.69% +/- 0.12 of the CD8^+^ T cells producing IFNγ eight days after immunization (Figure 2C). While the response was relatively small compared to the CD8^+^ T cell response stimulated from a live ZIKV infection, the response from the hAd5-ZKV was significantly above background.

The systemic antibody response was evaluated 21 days after vaccination. By 21 days post-immunization, it could be assumed that the majority of the vaccine-specific response, if it was generated, would have been immunoglobulin g (IgG) specific for the vaccine antigens. To measure the amount of ZIKV-specific antibody generated following vaccination, we performed a Western blot and an ELISA. To understand if the prM and E expressed by the hAd5-ZKV could be recognized by the polyclonal response after ZIKV infection, cell lysates from ZIKV- and hAd-ZKV-infected cells were probed with ZIKV polyclonal sera from wild type infected C57BL/6 mice (Figure 2D). The ZIKV immune serum detected the prM protein from the ZIKV-infected cells but not the hAd5-ZKV vaccine, potentially due to an increase in the cleavage of the pr peptide, which has been shown to block antibody recognition [48,57]. The expression of ZIKV E was evaluated by comparing E expression in ZIKV-infected versus hAd5-ZKV-infected cells. ZIKV E and hAd5-ZIKV E migrated at similar rates within the SDS-PAGE gel and were strongly recognized by the ZIKV immune serum. The fact that the hAd5-ZKV did not express NS1 is apparent with the NS1 multimeric ladder above ZIKV E in the infected cells, but it was not present in hAd5-ZKV vaccine-infected cells.

Based upon the recognition of the ZIKV E protein by Western blot after a single hAd5-ZKV vaccination, we next wanted to quantify the amount of anti-ZIKV E IgG by ELISA. Twenty-one days post vaccination, C57Bl/6 mice generated a ZIKV E-specific IgG response with a reciprocal mean endpoint titer of 2400 +/-1512 versus naïve animals with a reciprocal mean endpoint titer below the limit of detection of 25 (Figure 2E). To determine the quantity of neutralizing antibodies, focus reduction neutralization tests were completed, but the level of neutralizing antibodies was below the limit of detection for four out of six animals and was not significantly different compared to naïve animals (data not shown).

### 3.3. Immune Recognition by the Human Polyclonal Antibody Response

In our current understanding, prior natural ZIKV infection prevents the development of future ZIKV disease; one potential mechanism of that protection is sterilizing immunity [58,59]. To determine if the hAd5-ZKV vaccine generates antigens that are recognized by the human polyclonal antibody response that develops to naturally acquired ZIKV infection, we probed ZIKV- and hAd5-ZKV-infected cells with serum from humans who had tested positive for ZIKV infection by PCR (Figure 3A–D). We compared the human antibody response between a flavivirus-naïve individual where a response to hAd5 hexon was present, but antibody reactivity to the ZIKV proteins was absent. Three subjects that were identified as being ZIKV^+^ from 9–12 months post infection had polyclonal antibody responses that recognized the ZIKV E protein in both ZIKV- and hAd5-ZKV-infected cells. These data indicate that the hAd5-ZKV vaccine expressed ZIKV B cell epitopes that could be recognized by the human antibody response.

### 3.4. Protective Capacity of hAd5-ZKV Vaccine

To determine if the hAd5-ZKV vaccine could be protective, we administered 3 × 10^7^ viral particles i.n. of hAd5-ZIKV to *Ifnar1^-/-^* mice. Twenty-one days later, we lethally challenged the vaccinated animals and a PBS-vaccinated control group and monitored for survival (Figure 4A). After challenge, all control mice succumbed to ZIKV, while the majority of vaccinated mice (6/8) survived ZIKV challenge (*p* = 0.005) (Figure 4B). We scored both the weight loss (Figure 4C) and clinical signs of the disease (Figure 4D) for twelve days. All surviving vaccinated mice regained weight starting at day seven post infection and continued to gain weight for the duration of the experiment. In the evaluation of clinical scores, both control and vaccinated mice began to display symptoms of ZIKV infection on day six post infection. On day six post infection, the severity of the disease for the control mice continued until all mice became moribund by day eight, whereas during the same time period, 40% of the hAd5-ZKV-vaccinated mice showed no clinical signs of disease. Overall, these results demonstrate that the hAd5-ZIKV vaccine was able to induce a protective response leading to reduced disease severity in our animal model. Given that the mice were vaccinated once via the intranasal route, we found this strong evidence of T cell and antibody-mediated protection. Our study demonstrates the immunogenicity and protective capacity of hAd5-ZKV vaccine, as well as the presentation of ZIKV epitopes recognized by the human antibody response. Our findings demonstrate that the hAd5-ZKV vaccine warrants further safety and efficacy research.

## 4. Discussion

The development of a vaccine to protect against ZIKV disease is an urgent and currently unmet public health need. In 2014–2016, ZIKV demonstrated the ability to rapidly become an epidemic and cause devastating disease, putting tremendous stress on the global health system and on individuals. Though the incidence of ZIKV disease has diminished, the continued development of a ZIKV vaccine is required to prepare for the potential of another ZIKV epidemic, as well as for use as a traveler’s vaccine in areas where ZIKV transmission persists.

Here, we described the development of an hAd5 ZIKV vaccine that encodes and expresses the full length prM and E structural proteins from ZIKV virus PRVABC59. Flavivirus E proteins are known to be the a target of neutralizing antibodies and the T cell response [13,40,41]. The goal for including prM in this vaccine construct is to aid in the proper folding of Zika E proteins to allow for the proper folding of E and the presentation of neutralizing antibody epitopes, leading to a strong antibody response that has been shown in other similar vaccine constructs [60,61].

Our bioinformatic analysis and the work of others have shown that the prM and E sequences expressed by the hAd5-ZKV vaccine are highly conserved [62,63]. We demonstrated that a single i.n. vaccination with the hAd5-ZKV generates an antigen-specific CD8^+^ T cellular response and an IgG anti-ZIKV E response in wild type C57BL/6 mice. Lastly, using polyclonal sera from individuals who had naturally acquired ZIKV infection, we demonstrated that the antigens expressed by the hAd5-ZKV vaccine are recognized by the ZIKV-driven human polyclonal antibody response. Based upon our prior work as well as work by others, it can be said that the antigen-specific T cell and antibody responses can protect against ZIKV disease [19,21,22,64,65,66].

In order to determine the protective efficacy of this hAd5-ZKV, we used a *Ifnar1^-/-^* lethal infection model that has been used to validate several ZIKV vaccines currently in phase I trials [67,68]. Using this model, we demonstrated that a single hAd5-ZKV vaccination is sufficient for providing protection against a lethal challenge within *Ifnar1^-/-^* mice, as well as reducing weight-loss, improving clinical scores, and preventing mortality. Overall, our studies showed that the hAd5-ZKV vaccine can generate ZIKV antigens in a form that is recognized by the human polyclonal antibody response (Figure 3) and generate a protective immune response in a pre-clinical model of ZIKV disease (Figure 4).

The hAd5-ZKV vaccine candidate developed here included known human ZIKV T and B cell epitopes [69,70,71]. The inclusion of both T cell and B cell epitopes into this vaccine allowed for for the induction of the adaptive immune response and the potential for long-term memory. Virus-specific CD8^+^ T cells have the capacity to lyse virally-infected cells and produce cytokines, creating an environment that supports viral clearance and protection from severe disease (reviewed in [72]). For ZIKV, we showed that there is a broad CD8^+^ T cell response generated against the pathogen and that T cell response is required for protection against a lethal ZIKV challenge [19].

There have been a range of platforms used to develop ZIKV vaccines, including live attenuated and inactivated viruses, as well as protein subunit and viral vector-based vaccines (reviewed in [14,16]). Currently, two inactivated virus vaccines, two DNA vaccines, and one mRNA vaccine are being evaluated in phase I clinical trials [73]. For the experimental vaccination platforms of DNA and hAd5, replication-deficient hAd5 vectors have generated more robust human immune responses in comparison to DNA vaccinations for human immunodeficiency virus [74]. To our current knowledge, no clinical trial has directly compared human immune responses between replication-deficient hAd5 vaccines and a mRNA platform. The issue of hAd5 vectors as vaccine candidates has revolved around previous immunity to vector itself, leading to a minimized vaccine efficacy; however, this issue can be subverted using an intranasal (i.n.) route of vaccination [56]. Indeed, our study highlights the ability of a single i.n. vaccination with the hAd5-ZKV candidate described here to generate both T and B cell responses that are capable of protecting against severe ZIKV disease in a preclinical model of infection.

## Figures and Tables

**Figure 1 vaccines-08-00170-f001:**
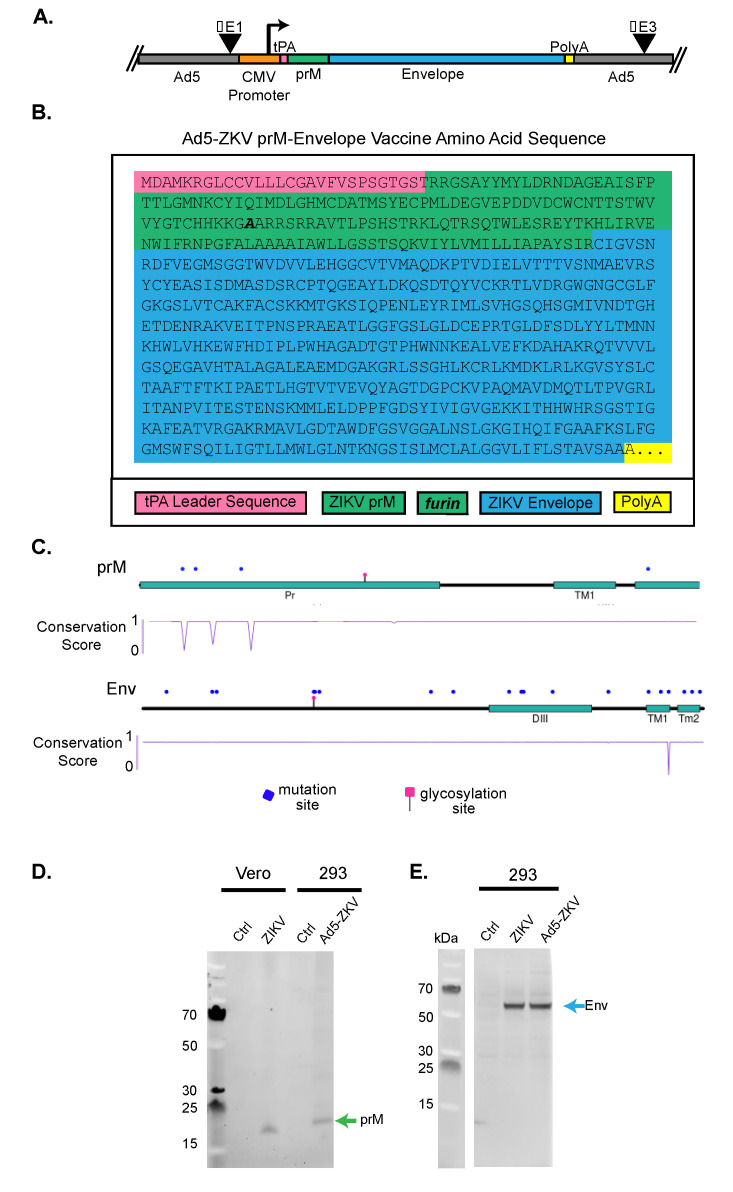
Development of human adenovirus type 5 Zika virus vaccine (Ad5-ZKV). (**A**) Schematic representation of the ZIKV structural protein expression cassette. hAd5 lacks the E1 and E3 coding regions, which are replaced with a cytomegalovirus (CMV) promoter-driven ZIKV cassette that expresses a codon-optimized tissue plasminogen activator (tPA) leader sequence followed by pre-membrane (prM) and full-length envelope (E) protein (hAd5-ZKV). (**B**) Expression cassette: The amino acid sequence of the tPA leader sequence, prM, and E based upon ZIKV strain PRVABC59 (ANK57897.1). The point mutation to increase furin cleavage is in bold. (**C**) Protein plot of prM and E. Using 249 ZIKV human isolate sequences, amino acid variations that occurred at a frequency above 0.08% (equal or greater than 2 variations at any particular site) were marked using a blue dot above the cartoon representation of the prM and E proteins. The cartoon representation consists of tracks highlighting key protein domains such as the pr portion of the prM and the DIII of the envelope. Below the cartoon representation is a track of the conservation score of the protein at each site that describes the level of identity between the ZIKV sequences based on amino acid variations. In this conservation plot, a score of 1 is representative of 100% amino acid identity, and a score of 0 is representative of 0% amino acid identity. (**D**) ZIKV prM protein expression from ZIKV and hAd5-ZKV: Vero cells were infected with ZIKV PRVABC-59, and 293F cells were infected with hAd5-ZKV, then cellular lysates were run on a SDS-PAGE non-reducing gel. prM was detected with rabbit anti-ZIKV prM (GTX-133305). (**E**) ZIKV E protein from ZIKV- and hAd5-ZKV-infected cells was detected with mouse anti-E 4G2.

**Figure 2 vaccines-08-00170-f002:**
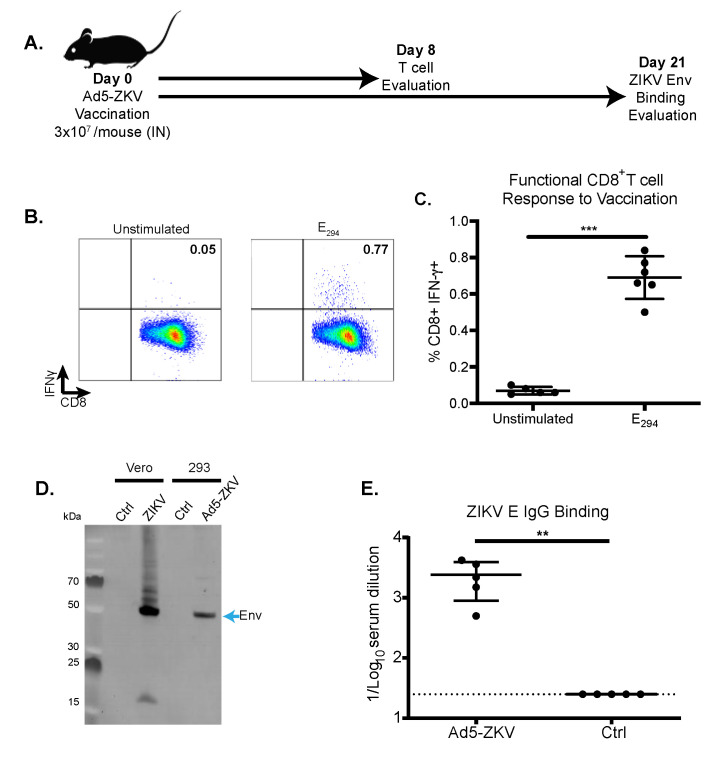
Immunogenicity of hAd5-ZKV vaccine in wild type C57BL/6 mice. (**A**) Graphical timeline for hAd5-ZKV vaccination and the evaluation of immunogenicity. C57Bl/6 mice were immunized with 3x10^7 viral particles of hAd5 intranasally (i.n.), and antigen-specific CD8^+^ T cells responses and antibody binding were evaluated. Antigen-specific T cell responses from the spleen of vaccinated mice were evaluated at day 8 post vaccination. (**B**) Representative flow plot of the intracellular interferon gamma production of CD8^+^T cells (gated on lymphocytes, CD8b^+^, and CD4^-^/CD19^-^) after ZIKV E_294_ peptide stimulation. (**C**) Percent of CD8^+^T cells secreting IFNγ in response to antigen-specific peptide stimulation with n = 6 per group. Asterisks indicate statistical significance * (*p* = 0.03), ** (*p* = 0.002), *** (*p* = 0.0002), and **** (*p* < 0.0001) as determined by Mann–Whitney test. **(D**) To determine the expression of ZIKV epitopes recognized after infection, ZIKV- and hAd5-ZKV-infected cells, and controls were evaluated by Western blot using ZIKV immune sera from C57Bl/6 mice. (**E**) Humoral responses were evaluated in the sera of immunized mice at day 21 post vaccination by ZIKV E protein binding with an n = 5 per group. Data are the cumulative results of two independent experiments, and significance was determined by student’s 2-tailed t test * (*p* = 0.03), ** (*p* = 0.002), *** (*p* = 0.0002), and **** (*p* < 0.0001).

**Figure 3 vaccines-08-00170-f003:**
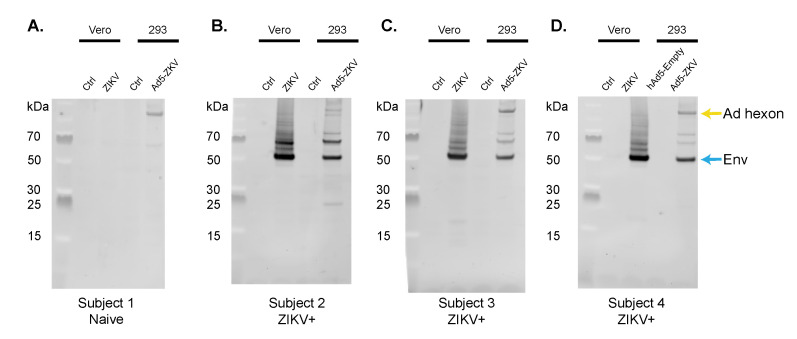
Recognition of hAd5-ZKV antigens by polyclonal human anti-ZIKV antibody response. Uninfected and ZIKV Vero cells, as well as uninfected and hAd5-ZKV-infected 293 cells, were probed with human serum from subjects that had a naturally acquired ZIKV infection, demonstrating recognition of ZIKV E antigens by the human immune response. (**A**) Serum from a flavivirus-naïve individual was used as a control, with staining for Ad5 hexon protein apparent. (**B**)–(**D**) Serum from three human positive subjects after 9–12 months after a ZIKV-positive test showed staining for E and hAd5 hexons.

**Figure 4 vaccines-08-00170-f004:**
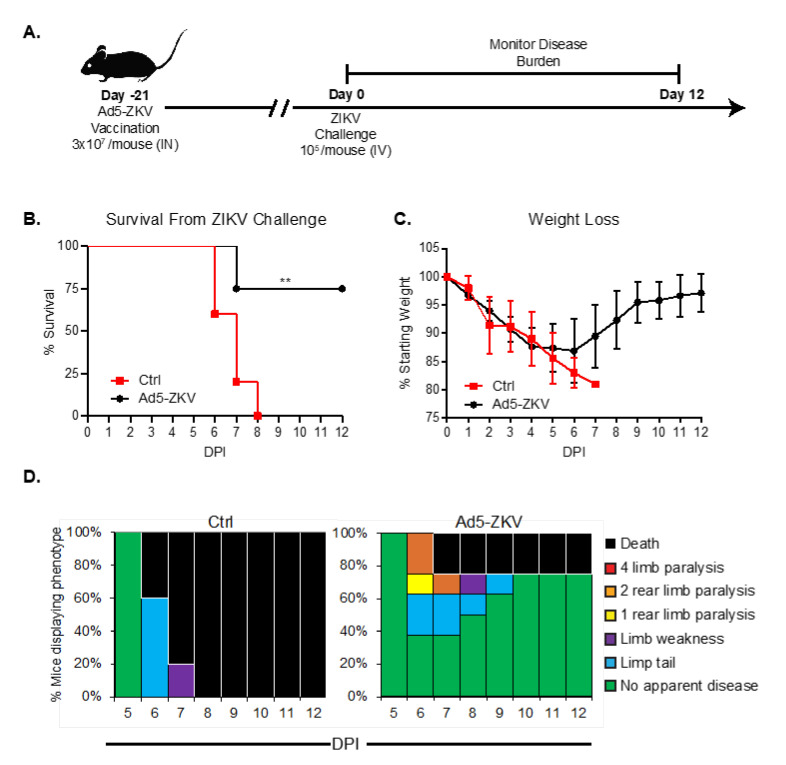
Protective Capacity of hAd5-ZKV vaccine vector encoding CD8^+^ T cell and B cell epitopes. (**A**) Graphical timeline for hAd5-ZKV vaccination. 3 × 10^7^ viral particles of hAd5-ZKV were administered intranasally to *Ifnar1^-/-^* mice. Twenty-one days later, both the vaccinated animals and a PBS-vaccinated control group were sub-lethally challenged with 1 × 10^4^ FFU i.v. of ZIKV PRVABC59, and weight loss was monitored. (**B**) ZIKV disease-associated clinical scores were monitored over the twelve-day period post challenge with n = 8 for vaccination and n = 5 for control. Statistical significance for survival (** *p* = 0.002) was determined using a log-rank (Mantel–Cox) test. (**C**) ZIKV-induced weight loss was monitored over the twelve-day period post challenge. Data are the cumulative results of two independent experiments. (**D**) Clinical disease score over the twelve-day period post challenge with ZIKV. Phenotypes evaluated included limp tail, limb weakness, limb paralysis, and death. Data are the cumulative results of two independent experiments.

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
