# Peer review of "Immunogenicity and Efficacy of a Recombinant Human Adenovirus Type 5 Vaccine against Zika Virus"

_vaccines, 2020, doi:10.3390/vaccines8020170_

Round 1
Reviewer 1 Report
In their manuscript entitled “Immunogenicity and efficacy of a recombinant human adenovirus type 5 vaccines against Zika virus,” Steffen and colleagues describe the successful development of an adenovirus-vectored Zika virus vaccine expressing Zika virus prM and E. This manuscript is well written and logically organized. The experiments appear to be scientifically valid and the conclusions sound. I have no major concerns with this manuscript, but I do think it would be helpful if the authors could expand their discussion of certain key aspects of the study and provide additional information, as outlined below.
1. The sequence analysis of prM and E is a bit confusing and could benefit from the inclusion of additional information. The legend for Fig 1(c) states that a blue dot indicates the presence of a “mutation” with a frequency of at least 0.08%. What was the range of frequencies detected in the analysis (in other words, were most of the amino acid changes detected at a low frequency or a high frequency)? Are changes that exist at a frequency of 0.08% relevant? Were the sites of variation (indicated by the blue dot) mostly the result of change to a specific amino acid, or were the changes more varied? Is it appropriate to refer to these changes as mutations, given that they may represent natural polymorphisms? Finally, how does the conservation plot relate to the dot plot? In some instances where an amino acid change is indicated by blue dot the conservation plot indicates a 1, which I assume to mean 100% identity? In other cases, the conservation plot indicates a value much less than 1, but there is no amino acid change indicated by a blue dot. Although I understand that these details do not have a direct impact on the efficacy of the vaccine, it would be helpful if these analyses were explained in a bit more detail.
2. What is the relative contribution of prM versus E to the overall immunogenicity/efficacy of the vaccine? The Ad5-expressed prM resolves differently on SDS-PAGE than ZIKV-expressed prM, and polyclonal sera from ZIKV-infected humans (Fig 3) does not seem to recognize prM. Did the authors examine the T cell response to a prM-derived epitope, or did they perform ELISAs using prM as the antigen? Does the altered expression of Ad5-expressed prM effect its immunogenicity, or is prM naturally less immunogenic? Is it necessary to include prM in the vaccine at all? An expanded discussion on these issues would be welcome.
3. Pursuant to the above point, Line 323 suggests that Fig 3 demonstrates recognition of “ZIKV prM and E antigens by the human immune response,” but prM does not appear to be detected in any case.
4. Can the authors discuss the issue of pre-existing immunity to Ad5-vectored vaccines, since this is a potential drawback to the further clinical development of this particular vaccine? Conversely, did the authors consider a prime/boost regimen to enhance the immune response? Would this have enhanced the immune response, or would the boost be negated immunity to the Ad5 vector?
5. On Lines 282-283, the authors state that the CD8+ T cell response to the vaccine was “relatively small compared to the CD8+ T cells response stimulated from a live ZIKV infection.” With respect to the live ZIKV infection, to what data are the authors referring? Is a citation missing?
6. The authors should indicate group size for each animal experiment.
7. The authors should include a description in the Materials and Methods (at least) of the statistical tests (and program) that were used to analyse the data and how the asterisks correspond to p values.
Author Response
Dear Editors and Reviewers,
We greatly appreciate your time and effort that you have invested in our manuscript. We have found all of
your questions, comments and criticisms extremely helpful. The comments and criticisms have helped us
develop a improved manuscript. We have submitted an updated copy of the manuscript where we have
tracked changes where we have addressed the reviewers. We have provided responses to the reviewers
in a point by point fashion below.
Reviewer #1
In their manuscript entitled “Immunogenicity and efficacy of a recombinant human adenovirus type 5
vaccines against Zika virus,” Steffen and colleagues describe the successful development of an
adenovirus-vectored Zika virus vaccine expressing Zika virus prM and E. This manuscript is well written
and logically organized. The experiments appear to be scientifically valid and the conclusions sound. I
have no major concerns with this manuscript, but I do think it would be helpful if the authors could expand
their discussion of certain key aspects of the study and provide additional information, as outlined below.
1. A. The sequence analysis of prM and E is a bit confusing and could benefit from the inclusion of
additional information. The legend for Fig 1(c) states that a blue dot indicates the presence of a “mutation”
with a frequency of at least 0.08%. What was the range of frequencies detected in the analysis (in other
words, were most of the amino acid changes detected at a low frequency or a high frequency)?
1a) We agree with the reviewer that this section deserved further explanation. There was a range of
frequencies detected in our sequence analysis, with 0-77 of the sequences varying from the vaccine in prM
to 0-6 of the sequences varying from the vaccine in envelope. Of note, the first three mutations we
highlighted in prM have low conservation scores, which relates to a high sequence variation within those
three amino acids between the 249 sequences we analyzed. However, a majority of the variations within E
were less frequent, making the conservation score much closer to 1, but not necessarily making a large
“dip” in the conservation score plot. To address the question specifically, a majority of the variations we
found were lower frequency overall (1-3 of the 249 sequences varied from the vaccine).
1b) Are changes that exist at a frequency of 0.08% relevant?
1b) This is a phenomenal question and a current active area of research in the field. For our purposes, a
frequency of 0.08% reflects equal or greater than 2 changes within in the particular amino acid. The blue
dots indicate changes at a particular amino acid. There is currently ongoing research in the field surrounding
viral genetic diversity in flaviviruses and impact on transmission and pathogenesis. Papers have included
mutations in prM that lead to congenital zika syndrome [Yuan et. al 2017], WNV adaptation to be transmitted
by North American Culex mosquitoes [Ebel et. al 2004], and mutations in NS1 leading to increased
infectivity in Aedes aegypti [Liu et. al 2017].
1c) Were the sites of variation (indicated by the blue dot) mostly the result of change to a specific amino
acid, or were the changes more varied? Is it appropriate to refer to these changes as mutations, given
that they may represent natural polymorphisms?
1c) This comment is very much so appreciated in the context of the figure. In our understanding the
delineating factor between mutation and polymorphism is the frequency within a population with a common
cutoff of 1%. That being said, some of the amino acids that we have highlighted could be categorized as
polymorphisms, for example the variation within the TM of ZIKV envelope. With this in mind, in the body of
our paper we refer to these changes in amino acids as “variations” in lieu of “mutation” or “polymorphism”,
but I have adjusted the wording in the figure legend that used “mutation.”
1d)Finally, how does the conservation plot relate to the dot plot? In some instances where an amino acid
change is indicated by blue dot the conservation plot indicates a 1, which I assume to mean 100%
identity? In other cases, the conservation plot indicates a value much less than 1, but there is no amino
acid change indicated by a blue dot. Although I understand that these details do not have a direct impact
on the efficacy of the vaccine, it would be helpful if these analyses were explained in a bit more detail.
1d) You are absolutely correct, indication of a 1 relates to 100% sequence identity and therefore no variation
in the sequence. However, a blue dot indicates a variation in the amino acid that would relate to a number
that is less than 1. For these variations represented by blue dots that do not look like there is a
corresponding dip in the conservation score it is nearly impossible to visualize in this way due to the low
level of sequence variation across the 249 sequences. For this reason, we opted for the additional blue
dots in concurrence with our conservation scores to try to accurately describe the natural variation. For
further clarification we have added all of the files related to the alignment as supplemental files.
2. a)What is the relative contribution of prM versus E to the overall immunogenicity/efficacy of the vaccine?
2a) These are all extremely thoughtful points of acknowledgement. In regard to efficacy of the vaccine, prM
is important for proper post translational folding of flavivirus E proteins, making it an essential part of this
vaccine as well as other similar constructs [Roby et, al 2005 and Allison et. al 1995]. As far as
immunogenicity, as it is technically a part of the structure of the virion it could be a target for the antibody
response but based on what we know about human antibody responses to a closely related flavivirus
dengue, we know that prM specific antibodies tend to be non-neutralizing and do not make up a bulk of the
dengue reactive antibodies [Tsai et. al 2017]. We have expanded the discussion to include these points.
2b) The Ad5-expressed prM resolves differently on SDS-PAGE than ZIKV-expressed prM, and polyclonal
sera from ZIKV-infected humans (Fig 3) does not seem to recognize prM.
2b) It is absolutely correct that there is a difference in the resolution of prM on SDS PAGE between ZIKV
infected cells and the hAd5 treated cells. We believe that this is most likely due to an alteration in cleavage
of the prM protein, but more experiments would need to be completed to determine the why and how and
are beyond the scope of this current manuscript. As previously stated, the human response is primarily
dominated by E specific antibodies, so it is not necessarily as shocking to us that there is little/no recognition
of prM by our human sera.
2c) Did the authors examine the T cell response to a prM-derived epitope, or did they perform ELISAs
using prM as the antigen? Does the altered expression of Ad5-expressed prM effect its immunogenicity,
or is prM naturally less immunogenic? Is it necessary to include prM in the vaccine at all? An expanded
discussion on these issues would be welcome.
2c) These are all wonderful and considerate points and based on this comment we have expanded on this
concept in the discussion section. We did not examine the T cell response or antibody response to prM
post vaccination. Our primary focus for protection was based on E, which has been shown in a mouse
model of infection to have the H2b restricted mouse immunodominant CD8+ epitope [Hassert et. al 2019].
We do not think that the altered expression of prM in our Ad5 vector has an effect on overall immunogenicity
as there was a detectable CD8+ and antibody response to the E protein. However, the addition of prM in
these types of constructs for flaviviruses is essential for the proper post-translational folding of E.
3. Pursuant to the above point, Line 323 suggests that Fig 3 demonstrates recognition of “ZIKV prM and E
antigens by the human immune response,” but prM does not appear to be detected in any case.
3) You are absolutely correct and identified an error in the description that we have modified in the
manuscript.
4. Can the authors discuss the issue of pre-existing immunity to Ad5-vectored vaccines, since this is a
potential drawback to the further clinical development of this particular vaccine? Conversely, did the
authors consider a prime/boost regimen to enhance the immune response? Would this have enhanced
the immune response, or would the boost be negated immunity to the Ad5 vector?
4) A potential barrier to use this kind of vector is that a proportion of the population has pre-existing immunity
against the hAd5 vector and there have been reports of immune interference associated with the
administration of these vectors parenterally. However, administration of Ad5 vectors using the i.n. route
allows for the expression of the protein of interest in the presence of pre-existing immunity to the vector
virus [Croyle 2008; Pandey 2012; Richardson 2013]. In relation to your second question, intranasal delivery
of an anthrax vaccine based on the same Ad5 vector technology in rabbits was associated with only modest
induction of anti-vector immunity and the immune response to the antigen was boosted in animals receiving
a two immunizations one-month apart [Krishnan et al.] This is a helpful suggestion from the reviewer and
we have further expanded on this point in the discussion.
5. On Lines 282-283, the authors state that the CD8+ T cell response to the vaccine was “relatively small
compared to the CD8+ T cells response stimulated from a live ZIKV infection.” With respect to the live
ZIKV infection, to what data are the authors referring? Is a citation missing?
5) Thank you for pointing out this deficiency, we were missing a citation that has been appropriately added.
6. The authors should indicate group size for each animal experiment.
6) We completely agree with this statement and indication of group size has been described in the figure
legend for each animal-based experiment.
7. The authors should include a description in the Materials and Methods (at least) of the statistical tests
(and program) that were used to analyse the data and how the asterisks correspond to p values.
7) We agree with the reviewer on the necessity of a section describing the statistical methods used
throughout this paper. We have added an additional section in the Materials and Methods describing all
statistical tests used in this paper. We have additionally added information indicating how each asterisk
corresponds to each P value into the corresponding figure legends.
References
1. Yuan, L.; Huang, X.-Y.; Liu, Z.-Y.; Zhang, F.; Zhu, X.-L.; Yu, J.-Y.; Ji, X.; Xu, Y.-P.; Li, G.; Li, C.,
et al. A single mutation in the prM protein of Zika virus contributes to fetal microcephaly. Science
2017, 358, 933-936, doi:10.1126/science.aam7120.
2. EBEL, G.D.; CARRICABURU, J.; YOUNG, D.; BERNARD, K.A.; KRAMER, L.D. GENETIC AND
PHENOTYPIC VARIATION OF WEST NILE VIRUS IN NEW YORK, 2000–2003. The American
Journal of Tropical Medicine and Hygiene 2004, 71, 493-500,
doi:https://doi.org/10.4269/ajtmh.2004.71.493.
3. Liu, Y.; Liu, J.; Du, S.; Shan, C.; Nie, K.; Zhang, R.; Li, X.-F.; Zhang, R.; Wang, T.; Qin, C.-F., et
al. Evolutionary enhancement of Zika virus infectivity in Aedes aegypti mosquitoes. Nature 2017,
545, 482-486, doi:10.1038/nature22365.
4. Roby, J.A.; Setoh, Y.X.; Hall, R.A.; Khromykh, A.A. Post-translational regulation and
modifications of flavivirus structural proteins. Journal of General Virology 2015, 96, 1551-1569,
doi:https://doi.org/10.1099/vir.0.000097.
5. Allison, S.L.; Stadler, K.; Mandl, C.W.; Kunz, C.; Heinz, F.X. Synthesis and secretion of
recombinant tick-borne encephalitis virus protein E in soluble and particulate form. Journal of
Virology 1995, 69, 5816-5820.
6. Tsai, W.-Y.; Lin, H.-E.; Wang, W.-K. Complexity of Human Antibody Response to Dengue Virus:
Implication for Vaccine Development. Front Microbiol 2017, 8, 1372-1372,
doi:10.3389/fmicb.2017.01372.
7. Hassert, M.; Harris, M.G.; Brien, J.D.; Pinto, A.K. Identification of Protective CD8 T Cell
Responses in a Mouse Model of Zika Virus Infection. Front Immunol 2019, 10, 1678-1678,
doi:10.3389/fimmu.2019.01678.
8. Croyle, M.A.; Patel, A.; Tran, K.N.; Gray, M.; Zhang, Y.; Strong, J.E.; Feldmann, H.; Kobinger,
G.P. Nasal delivery of an adenovirus-based vaccine bypasses pre-existing immunity to the
vaccine carrier and improves the immune response in mice. PLoS One 2008, 3, e3548-e3548,
doi:10.1371/journal.pone.0003548.
9. Pandey, A.; Singh, N.; Vemula, S.V.; Couëtil, L.; Katz, J.M.; Donis, R.; Sambhara, S.; Mittal, S.K.
Impact of preexisting adenovirus vector immunity on immunogenicity and protection conferred
with an adenovirus-based H5N1 influenza vaccine. PLoS One 2012, 7, e33428-e33428,
doi:10.1371/journal.pone.0033428.
10. Richardson, J.S.; Pillet, S.; Bello, A.J.; Kobinger, G.P. Airway delivery of an adenovirus-based
Ebola virus vaccine bypasses existing immunity to homologous adenovirus in nonhuman
primates. Journal of virology 2013, 87, 3668-3677, doi:10.1128/JVI.02864-12.
11. Krishnan, V.; Andersen, B.H.; Shoemaker, C.; Sivko, G.S.; Tordoff, K.P.; Stark, G.V.; Zhang, J.;
Feng, T.; Duchars, M.; Roberts, M.S. Efficacy and immunogenicity of single-dose AdVAV
intranasal anthrax vaccine compared to anthrax vaccine absorbed in an aerosolized spore rabbit
challenge model. Clin Vaccine Immunol 2015, 22, 430-439, doi:10.1128/CVI.00690-14.
Reviewer 2 Report
The authors present a well-written and technically sound paper on the development of an adenovirus vector-based ZIKV vaccine, with in vitro and in vivo evidence of immunogenicity and protection. I have only a few minor comments:
1) Line 70: missing bracket
2) Understanding that the intranasal administration of the vaccine bypasses pre-existing immunity to the vector, have the authors assessed immunogenicity and protection of their vaccine via an injection route? If so, this could be phrased in the discussion.
3) Line 148: Amersham and Typhoon should be capitalized.
4) Fig. 1d): Is the effect of the different band molecular weights consistent between replicate gels of this experiment? It looks to me that the gel might have run a bit crooked in this particular panel. Have the authors tried cutting out the bands and assessing the protein identity using mass spectrometry to make sure it's the same protein between cell lines?
5) Line 268: comma after "infection"
6) Line 269: "was" should be "were"
7) Line 271: "is" should be "are"
8) Line 352: "is" should be "are"
Author Response
Dear Editors and Reviewers,
We greatly appreciate your time and effort that you have invested in our manuscript. We have found all of
your questions, comments and criticisms extremely helpful. The comments and criticisms have helped us
develop a improved manuscript. We have submitted an updated copy of the manuscript where we have
tracked changes where we have addressed the reviewers. We have provided responses to the reviewers
in a point by point fashion below.
Reviewer 2:
Comments and Suggestions for Authors
The authors present a well-written and technically sound paper on the development of an adenovirus
vector-based ZIKV vaccine, with in vitro and in vivo evidence of immunogenicity and protection. I have
only a few minor comments:
1) Line 70: missing bracket
1)We have added this omitted bracket on line 70.
2) Understanding that the intranasal administration of the vaccine bypasses pre-existing immunity to the
vector, have the authors assessed immunogenicity and protection of their vaccine via an injection route?
If so, this could be phrased in the discussion.
2)We have not assessed immunogenicity of this vaccine via any route besides intranasal, however because
of the importance of this comment, we expanded on this topic in the discussion. We initially chose this
injection route based on a particular paper using an Ad5 expressing Ebola vaccine [Croyle et. al 2008]. This
paper highlights that i.n. vaccination has higher immunogenicity over i.m. and oral routes. Additionally, this
paper highlighted that only mice vaccinated i.n. in the presence of pre-existing immunity survived.
3) Line 148: Amersham and Typhoon should be capitalized.
3)We thank this reviewer for realizing a misstep on our part in our revision process and the grammatical
corrections have been made.
4) Fig. 1d): Is the effect of the different band molecular weights consistent between replicate gels of this
experiment? It looks to me that the gel might have run a bit crooked in this particular panel. Have the
authors tried cutting out the bands and assessing the protein identity using mass spectrometry to make
sure it's the same protein between cell lines?
4)We did not anticipate seeing this difference in banding of prM, however we do believe that this
phenotype is due to an altered cleavage of prM from E and is consistent between gels. We have not
performed any mass spectrometry on these bands, but since detection on this particular blot was done
with an anti-prM antibody we are confident that it is our protein of interest.
5) Line 268: comma after "infection"
6) Line 269: "was" should be "were"
7) Line 271: "is" should be "are"
8) Line 352: "is" should be "are"
5)-8)This response is for comments 5-8 due to their similarity in nature. We have corrected all grammatical
errors pointed out by the reviewer.
References
Croyle, M.A.; Patel, A.; Tran, K.N.; Gray, M.; Zhang, Y.; Strong, J.E.; Feldmann, H.; Kobinger, G.P. Nasal
delivery of an adenovirus-based vaccine bypasses pre-existing immunity to the vaccine carrier and
improves the immune response in mice. PLoS One 2008, 3, e3548-e3548,
doi:10.1371/journal.pone.0003548.